# Remarkable Alteration of PD-L1 Expression after Immune Checkpoint Therapy in Patients with Non-Small-Cell Lung Cancer: Two Autopsy Case Reports

**DOI:** 10.3390/ijms20102578

**Published:** 2019-05-26

**Authors:** Toshiaki Takahashi, Akiko Tateishi, Andrey Bychkov, Junya Fukuoka

**Affiliations:** 1Department of Postgraduate Medical Education, Kameda Medical Center, Kamogawa, Chiba 296-8602, Japan; markgreene00@gmail.com; 2Department of Pulmonology, Kameda Medical Center, Kamogawa, Chiba 296-8602, Japan; akikotoku1002@gmail.com; 3Department of Pathology, Kameda Medical Center, Kamogawa, Chiba 296-8602, Japan; bychkov.andrey@kameda.jp; 4Department of Pathology, Nagasaki University Graduate School of Biomedical Sciences, Nagasaki 852-8523, Japan

**Keywords:** NSCLC, immune therapy, Pembrolizumab, immune checkpoint inhibitors, PD-L1, tumor microenvironment

## Abstract

Pembrolizumab is an immune checkpoint inhibitor (ICI), currently recommended as the first-line treatment for patients with advanced non-small-cell lung cancer (NSCLC) showing ≥50% expression of programmed death-ligand 1 (PD-L1). Previously it was reported that platinum-based chemotherapy may change PD-L1 expression in solid cancers. However, no reports addressing alteration of PD-L1 expression after ICI therapy in NSCLC are available so far. The patients were Japanese males 83 and 87 years old, who were diagnosed with NSCLC based on the transbronchial lung biopsies showing sarcomatoid feature with high PD-L1 expression. They received Pembrolizumab, however, passed away with disease progression on day 27 and day 9, respectively. PD-L1, PD1, and CD8 antibodies were applied to pretreatment tumor biopsies and autopsy specimens. Immunoexpression of all the markers was evaluated using Aperio ImageScope. We found that PD-L1 expression decreased significantly from 75.6% to 13.2% and from 100% to 58.8%, in patients 1 and 2, respectively. This alteration was less prominent in the perinecrotic tumor area. A considerable decrease of PD-L1 score was linked with a little effect of Pembrolizumab in our patients. This association might be one of the contributing mechanisms of resistance to ICI and needs further investigation in large-scale studies.

## 1. Introduction

Lung cancer is the leading cause of cancer-related death worldwide. Non-small-cell lung cancer (NSCLC) accounts for approximately 80% of all lung cancer cases [1]. Introduction of immune checkpoint inhibitor (ICI) therapy significantly improved progression-free and overall survival of patients with NSCLC compared with conventional platinum-based chemotherapy. ICIs, such as Pembrolizumab, are currently recommended as frontline treatment in patients with advanced epidermal growth factor receptor (*EGFR*) mutations/anaplastic lymphoma kinase (*ALK*) gene fusions, ROS proto-oncogene 1 (*ROS1*) gene fusions, and B-Raf proto-oncogene (*BRAF*) gene mutations wild-type NSCLC and programmed death-ligand 1 (PD-L1) tumor proportion score ≥50% [2,3]. Programmed cell death protein 1 (PD-1) is one of the coinhibitory receptors expressed on the surface of antigen-stimulated T cells. PD-L1 and PD-L2 are its ligands on the surface of tumor cells [4].

Numerous reports on lung and other solid cancers showed that PD-L1 expression can be altered after platinum-based chemotherapy or concurrent chemoradiation [5]. In a representative series, Fujimoto et al. showed that PD-L1 expression significantly decreased after the concurrent chemoradiation therapy in locally advanced non-small lung cancer, which was associated with favorable prognosis [6]. In the trial, the median overall survival of patients with decreased, unchanged, or increased PD-L1 expression was 85.1, 92.8, and 14.6 months, respectively [6]. Interestingly, despite direct link between PD-L1 and ICI, no reports on dynamics of PD-L1 expression after ICI therapy in lung cancer are available.

We present two patients of NSCLC with fatal outcome, which failed to respond to Pembrolizumab. Both cases showed altered PD-L1 expression of tumor cells and immune cells distribution after ICI administration. We speculate that there is a possible association between the alteration of PD-L1 expression and ICI treatment outcome in patients with NSCLC.

## 2. Results

### 2.1. Case 1

An 83-year-old male with a 40 pack-year smoking status and no relevant medical history was admitted to our hospital presenting exertional dyspnea for two weeks. He also complained about his right upper back pain and unintentional weight loss. On admission, his performance status, according to the Eastern Cooperative Oncology Group, was 3, and his vital signs were normal and physical examination was remarkable for decreased breath sounds on the right side of the chest. Contrast computed tomography (CT) scan showed right contrast-enhanced pleural thickness with massive pleural effusion (Figure 1A,B). 18F-fluorodeoxyglucose-positron emission tomography (FDG-PET) showed high FDG uptake in the thickened right pleura and mediastinal and cervical lymph nodes (Figure 1C,D). He was suspected of having malignant mesothelioma and received CT-guided pleuropulmonary biopsy. Pathological diagnosis based on microscopic and immunohistochemical findings was poorly differentiated non-small-cell carcinoma with sarcomatoid differentiation (Figure 1E,F). The clinical stage as per the 8th edition AJCC/TNM was T4N3M1c (stage IVB). Molecular studies detected no *ALK* rearrangement and *EGFR* mutation. Immunostaining with anti-PD-L1 revealed high PD-L1 expression; a tumor proportion score (TPS) after the manual evaluation was reported as 65%.

The patient was treated with ICI Pembrolizumab (200 mg per course/body). At day 8, his white blood cell count increased up to 36,300/μL. His respiratory status was initially improved but his condition gradually got worse. At day 15, chest CT revealed increased circumferential thickness of right pleura and increased amount of pleural effusion. His status was considered as progressive disease according to the Response Evaluation Criteria in Solid Tumors (RECIST) criteria and the next administration of Pembrolizumab was postponed. The patient passed away at day 28 due to multiple organ failure. Postmortem CT showed lobular consolidation in both lungs (Figure 1G,H). Autopsy revealed medullary variegated hemorrhagic–necrotic cancer encasing the whole right lung, suggesting the pseudomesotheliomatous lung cancer, with metastasis to lymph nodes, adrenal glands, and vertebral column.

### 2.2. Case 2

An 86-year-old male, who had 60 pack-year smoking status and no relevant medical history, was admitted to our hospital presenting hematochezia. His performance status was 3 and his vital signs and physical exam were unremarkable. CT detected a mass lesion in S6 of the right lung (Figure 2A,B). FDG-PET scan showed high tracer uptake in the right hilar region and in the liver (Figure 2C,D), suggesting local progression and systemic metastasis. Transbronchial biopsy revealed poorly differentiated squamous cell carcinoma positive for TTF-1 and negative for p40. The patient was diagnosed as squamous cell carcinoma (cT2aN0M0, stage IB) and underwent lobectomy of the right lower lobe with mediastinal lymph node dissection. Histopathological examination of the surgical specimen showed spindle cells and giant cells (Figure 2E,F), which was consistent with pleomorphic carcinoma with no evidence of lymph node metastasis (pT3N0M0, stage IIB). The tumor was negative for *ALK* rearrangements, and *EGFR* and *ROS-1* mutations. No adjuvant chemotherapy was administered.

Three months after the surgery, PET-CT revealed local recurrence and systemic metastases at the follow-up visit. Additional immunostaining of the surgical tumor specimen showed high PD-L1 expression with 90% TPS after manual evaluation, and therefore the patient was treated with Pembrolizumab (200 mg/body). During the treatment, his white blood cell count elevated up to 61,100/μL on day 3. He passed away on day 9 due to respiratory failure. Postmortem CT showed the right bronchial invasion of the tumor causing the collapse of the right lung and the massive right pleural effusion (Figure 2G,H). An autopsy revealed the local recurrence of the carcinoma involving the hilar area of the right upper lobe. The cancer spread to the right adrenal, liver, and paraaortic abdominal lymph nodes.

### 2.3. Immunohistochemical Findings

In both cases, pathological findings were remarkable for the altered PD-L1 expression that turned from high into low to moderate (Figure 3). Digital image analysis revealed that PD-L1 expression significantly decreased in both cases, particularly in the viable tumor area (Table 1). PD-1 switched from completely negative expression to occasionally positive. CD8-positive cells tended to infiltrate more in the viable tumor area after ICI course. However, there was an opposite trend in the perinecrotic area.

## 3. Discussion

In this study, we found the expression of PD-L1 on tumor cells turned from high to low and the trend that viable cells showed less expression than necrotic cells. Evaluation of biopsy samples after the ICI therapy is not a common practice, and we believe this is the first study to demonstrate the alteration of PD-L1 expression after the ICI therapy. This investigation showed the possibility that the alteration of the PD-L1 expression is associated with the failure of ICI therapy.

Mechanism of action of ICI is associated with the reactivation and clonal proliferation of antigen-exposed T cells [7]. Antigen-presenting cells, such as dendritic cells and B cells, present tumor-associated peptide antigens and T cell receptors of naïve T cells recognize these peptides displayed by major histocompatibility complex, which initiate activation of T cells. Blockade of the PD-1 signaling axis prevents PD-1-mediated attenuation of proximal T cell receptor signaling, allowing for restoration of activity of immune cells [8].

In our two autopsy cases, the expression of PD-L1 decreased remarkably after the ICI administration (Figure 3), which could be related to the potential mechanism of failure of the ICI therapy. The mechanism of resistance to anti-PD-1/PD-L1 therapy is not fully understood. Several contributing factors including tumor neoantigen expression, the switch of cellular signaling pathways, tumor microenvironment (TME), and epigenetic modification were reported [5]. Even if tumor-specific antigens are presented on the surface of the cancer cells, ICI does not work effectively if the function of the activated T cells is inadequate or exhausted [9]. Particularly, failure of ICI therapy can be associated with inadequate generation of antitumor T cells, inadequate functions of activated T cells, and impaired function of T cell memory [7].

The TME also plays a key role in the efficacy of ICI therapy. Various factors are involved in the TME. Major players include myeloid cells, cytokines, Foxp3^+^CD4^+^ regulatory T cells, and cancer-associated fibroblasts [10]. Regulatory T cells exert an inhibitory effect by inducing various cytokines and directly inhibiting T cell proliferation. The response to ICI therapy was associated with the increased ratio of effector T cells to regulatory T cells [11]. In terms of the TME, one of the important mechanisms of resistance is upregulation of PD-1 ligands in the TME and connection of the PD-L1 to CD8^+^ T cells [11]. Encountering tumor antigen prompts infiltrated CD8-positive T cells to secrete IFN-γ which leads to the upregulation of PD-L1 in tumor cells [12].

In our cases, a significant decrease in PD-L1 expression was accompanied by tumor progression in a short period. Contrast CT showed that the tumor volume increased several times in both cases (Figure 1A,G and Figure 2A,G), thus various mechanisms such as low response of ICI therapy, the exhaustion of T cells, and the high rate of tumor proliferation based on the histological feature (pleomorphic feature) could be hypothesized.

There are a few studies which reported the dynamics of PD-L1 expression by cancer cells in patients with NSCLC treated with chemotherapy. Fujimoto et al. found that decrease of the PD-L1 expression after the concurrent chemoradiation therapy was related to the favorable prognosis [6]. Sheng et al. also showed the negative-to-positive switch of PD-L1 status was significantly associated with poor prognosis [13]. Another study reported a significant decrease of PD-L1 expression in patients with NSCLC treated with cisplatin-based chemotherapy but did not evaluate any correlation with outcome [5]. On the other hand, two independent Japanese groups demonstrated the opposite results. Omori et al. found that patients with NSCLC treated with *EGFR* tyrosine kinase inhibitors showed increase in PD-L1 expression, though no obvious correlation of increase or decrease of PD-L1 expression and survival was drawn [14]. The most recent study reported that PD-L1 expression tended to be increased in rebiopsy samples compared with the initial biopsy in patients with NSCLC wild-type for *EGFR* mutations and *ALK* rearrangement who underwent cytotoxic chemotherapy [15]. In summary, existing data on the change of PD-L1 expression in response to chemotherapy are conflicting, and there has been no reported clear association between such alteration and prognosis.

The issue of the change of PD-L1 expression on tumor cells due to ICI therapy has not been addressed so far. Intuitively, we would assume that PD-L1 expression remains high or unchanged in patients who responded to the ICI therapy, and it may likely decrease in the patients who failed the therapy. As it is recommended that ICI is administered to the NSCLC patients with PD-L1 expression over 50%, the more PD-L1 expressed on tumor cells, the more effective the ICI response [3]. It was shown that the lower PD-L1 expression is linked with the less efficient ICI therapy [16]. Therefore, a dynamic decrease of PD-L1 expression during ICI treatment might be considered to negatively correlate with the treatment success.

In order to evaluate PD-L1 expression in the context of the TME, we focused on two geographic zones, including viable tumor area and perinecrotic area. We observed the alteration of PD-L1 expression in both zones. The area populated with survived tumor cells showing relatively lower expression of PD-L1 was thought to be “resistant” to ICI therapy, while the perinecrotic area showing higher PD-L1 expression compared to the viable area was thought to be “sensitive” to ICI therapy.

One of the reasons for alteration in both areas might be less stimulation of PD-1 to PD-L1 which downregulates PD-L1 expression. To address TME-related perturbations, immunostaining with PD-1 and CD8 antibodies was performed. PD-1 expression slightly increased but such change might not be significant. One of the possible contributing mechanisms is that PD-1 molecule was “masked” due to binding between PD-1 and PD-1 inhibitor. Percentage of CD8^+^ T cells slightly increased in the viable tumor area. Theoretically, CD8^+^ T cells are prompted to infiltrate into the tumor interface, and it was reported that PD-1 inhibitor increased the intratumor CD8/Treg ratio [17]. In the murine model of breast cancer, neoadjuvant ICI therapy resulted in a sustained increase of tumor-specific CD8^+^ T cells in the blood, which predicted long-term survival [18]. In summary, changes in PD-1 expression and infiltration by CD8 cells were rather insignificant compared to the dynamics of PD-L1.

This study has some limitations which need to be considered in the future larger-scale studies. For instance, heterogeneity of PD-L1 expression in cancer cells, which is caused by the polyclonal evolution of the tumor [19] and also influenced by TME [20], may be challenging for scoring. In the same sense, an autopsy sample of the tumor may not exactly match an area of tumor tissue obtained by biopsy before the ICI therapy, which can be considered as an inherent limitation. PD-L1 heterogeneity often creates difficulties in the evaluation, which lays a foundation for interobserver variation. To overcome this obstacle, we employed an automated digital image analysis. It was not surprising to find that the manual score of PD-L1 expression showed a certain discordance (about 10% lower TPS) compared to the automated scoring. There is a current trend to implement digital image analysis in the workflow of PD-L1 protocols [21].

## 4. Materials and Methods

### 4.1. Immunohistochemistry

Immunostaining was performed on formalin-fixed, paraffin-embedded whole tissue sections using antibodies specific to PD-L1 (clone 22C3; prediluted, Agilent, Palo Alto, CA, USA), PD-1 (1:50; Abcam, Cambridge, UK), and CD8 (clone C8/144B; prediluted, Nichirei, Tokyo, Japan). Tissue blocks containing the most representative and well-preserved tumor areas were selected for immunostaining. Four-micrometer-thick sections were positioned on positively charged slides, dewaxed in xylene, and rehydrated using graded alcohols. The whole staining procedure from blocking to counterstaining was performed on Dako Autostainer Link 48 (Dako, Carpinteria, CA, USA) according to the manufacturers’ protocols.

### 4.2. Evaluation of Immunostaining and Digital Image Analysis

All slides stained with the antibodies to PD-L1, PD-1, and CD8 were digitized by MoticEasyScan digital slide scanner (Motic, Hong Kong, China) on −40 magnification. Image analysis was performed using the Aperio Imagescope v.12.1.0.5029 software package (Aperio Technologies, Inc., Vista, CA, USA). Immunostaining was automatically scored using the commercially available Nuclear v.9 and Membranous v.9 algorithms (Aperio Technologies, Inc., USA). The membranous algorithm was applied to score PD-L1 expression and the nuclear algorithm was used for PD-1 and CD8 immunostains. PD-L1 was evaluated in all cancer cells and the tumor proportion score was defined as the percentage of viable tumor cells showing partial or complete membrane staining (≥1%) relative to all viable tumor cells present in the sample (positive and negative) [22]. PD-1 and CD8 were scored in hot spots. The quantification of the CD8^+^ T cells was accomplished by calculating the number of the CD8^+^ T cells per area of 0.6 mm^2^.

Expression of the target proteins was analyzed in two areas of the tumor. Major was the area occupied by the viable cancer cells (i.e., resistant to ICI therapy). In addition, we separately evaluated the perinecrotic area, which is supposed to represent alive but degenerating cancer cells.

## 5. Conclusions

We reported two autopsy cases with dynamic alteration of PD-L1 expression after administering ICI. This observation might be one of the contributing mechanisms to the ICI resistance. Currently there are numerous ongoing clinical trials of neoadjuvant ICI therapy in various malignancies, however, there have been no reports on the correlation between alteration of PD-L1 expression after ICI therapy and its outcome. We believe that monitoring of PD-L1 expression during the treatment to seek the better outcome of the ICI therapy deserves further detailed investigation.

## Figures and Tables

**Figure 1 ijms-20-02578-f001:**
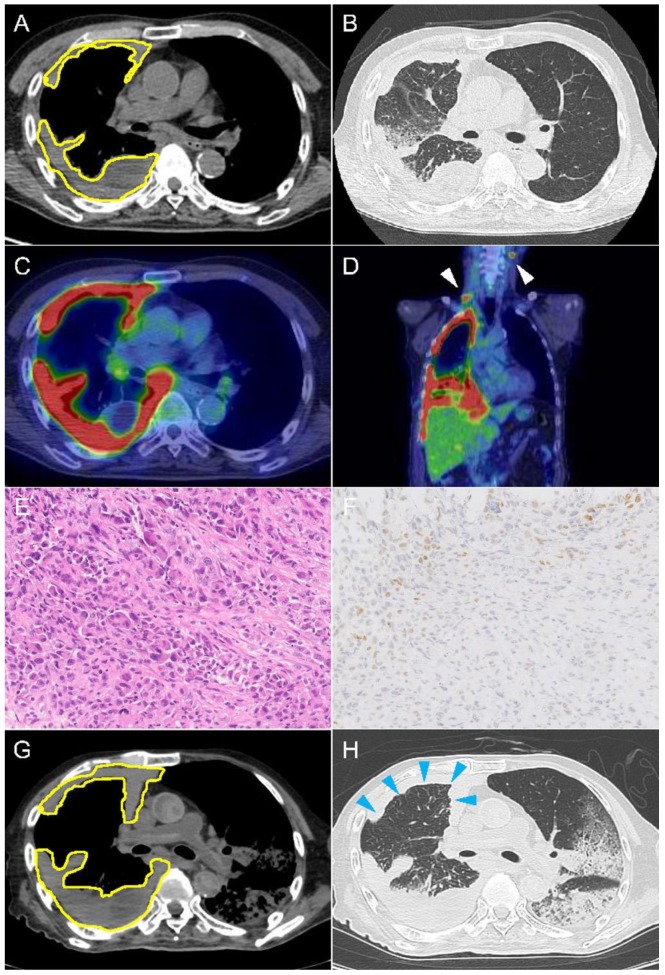
Imaging and histopathological findings in case 1. Chest computed tomography showing tumor (yellow outline) before ICI treatment on mediastinal (**A**) and lung (**B**) window images. PET-CT scan found high FDG uptake in the thickened right pleura (**C**), mediastinal and cervical lymph nodes (**D**, white arrowhead). Hematoxylin & eosin staining revealed poorly differentiated carcinoma, 200× (**E**). Immunostaining with TTF-1 demonstrated only focal residual expression, 200× (**F**). Postmortem chest tomography showed significantly increased circumferential pleural thickness (blue arrowhead) on mediastinal (**G**) and lung (**H**) window images.

**Figure 2 ijms-20-02578-f002:**
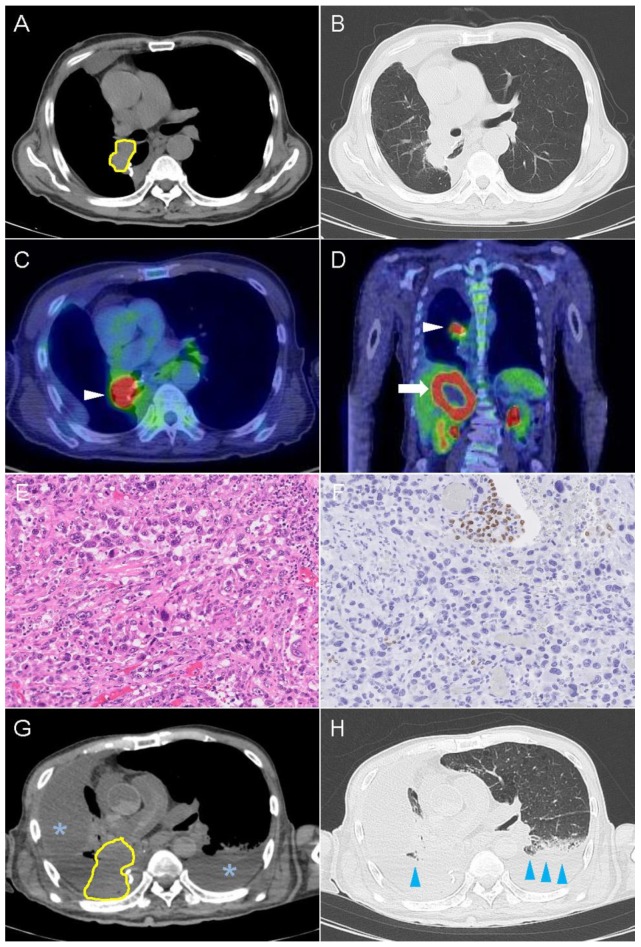
Imaging and microscopic findings in case 2. Computed tomography scan of the chest showing tumor (yellow outline) before ICI treatment on mediastinal (**A**) and lung (**B**) window images. PET-CT scan found high FDG uptake in the mass located in right hilar region (**C**, white arrowhead) and in the liver (**D**, white arrow). Routine hematoxylin & eosin staining revealed pleomorphic carcinoma with giant and spindle cells, 200× (**E**). Immunostaining with TTF-1 demonstrated loss of expression in the most carcinoma cells and residual expression in the entrapped bronchial and alveolar epithelium, 200× (**F**). Postmortem chest scan showed increased amount of pleural effusion (asterisk) and bilateral consolidation (blue arrowhead) on mediastinal (**G**) and lung (**H**) window images.

**Figure 3 ijms-20-02578-f003:**
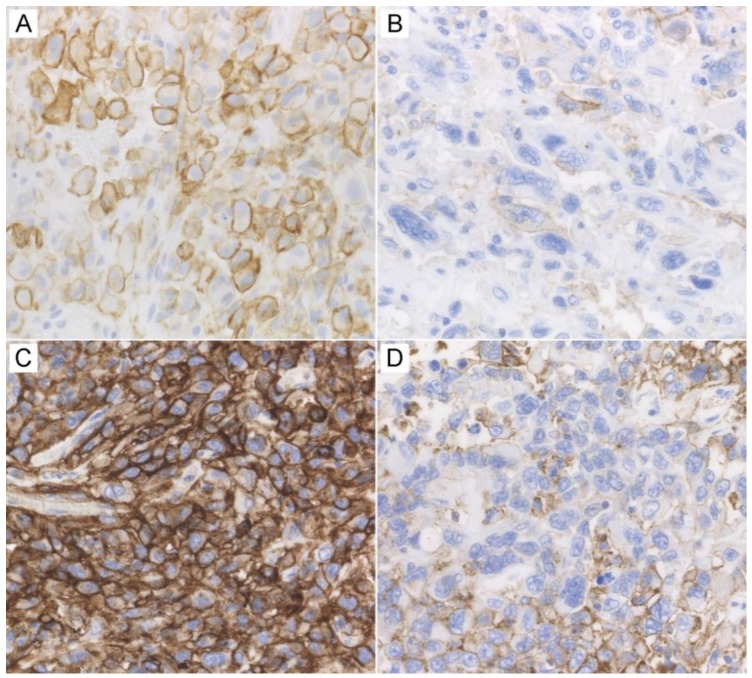
Dynamics of PD-L1 expression. High expression of PD-L1 on cancer cells in patients 1 (**A**) and 2 (**C**), which allowed administration of Pembrolizumab. A remarkable decrease of PD-L1 expression in patients 1 (**B**) and 2 (**D**) after immunotherapy. Immunohistochemistry, ×400 (**A**–**D**).

**Table 1 ijms-20-02578-t001:** Immunohistochemical expression of PD-L1 and selected biomarkers in non-small-cell lung carcinoma specimensbefore and after immunotherapy.

Cases	Markers	Before Immunotherapy	After Immunotherapy	
			Viable Tumor Area	Perinecrotic Area
Patient 1	PD-L1	75.6%	13.2%	31.7%
	PD-1	0.0%	0.7%	0.3%
	CD8	6.3	10.1	6.6
Patient 2	PD-L1	100.0%	58.8%	79.7%
	PD-1	0.0%	4.5%	1.2%
	CD8	21.5	28.8	4.4

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
