# Peer review of "Remarkable Alteration of PD-L1 Expression after Immune Checkpoint Therapy in Patients with Non-Small-Cell Lung Cancer: Two Autopsy Case Reports"

_ijms, 2019, doi:10.3390/ijms20102578_

Reviewer 1 Report

Overall, the two cases reported in the manuscript are very invaluable in clinical point of view, although the authors didn’t fully elucidate the deep mechanism and the data is not sufficient enough to draw any deep conclusions. It is worth to publish as a case report article as it still provides some important information for other clinicians as reference. Here are few minor points and would like to see the clarification/amendments by the authors:

1. Manuscript mentioned that “At day 15, chest CT revealed increased circumferential thickness of right pleura and increased amount of pleural effusion.” ”Three months after the surgery, PET-CT revealed local recurrence and systemic metastases at the follow-up visit.”18F-fluorodeoxyglucose-positron emission tomography (FDG- PET) showed high FDG uptake in the thickened right pleura, mediastinal and cervical lymph nodes.” These results are valuable to demonstrate the changed status of patients. The images of these tomography should be added in the manuscript.

2. Manuscript also mentioned that “Pathological diagnosis based on microscopic and immunohistochemical findings was poorly differentiated non-small cell carcinoma with sarcomatoid differentiation.” “Histopathological examination of the surgical specimen showed spindle cells and giant cells, which was consistent with pleomorphic carcinoma with no evidence of lymph node metastasis”. The images of immunohistochemical result and Histopathological examination also should be added in the manuscript.

3. Manuscript mentioned that “Contrast CT showed that the tumor diameter increased five times in case 1 and two times in case 2 (Fig. 1–2)”. Tumor should be marked by labels, such as arrows, to make the changed tumor diameter easy to show.

4.” To address TME-related perturbations, immunostaining with PD-1 and CD8 antibodies was performed.” What is reason for detecting CD8+ T cell status? How about detecting the status of CD4+ T cell or CD3+ T cell? Previous studies also mentioned that “ICI therapy resulted in a sustained increase of tumor-specific CD8+ T cells in the blood, which predicted long-term survivalIn the manuscript, Have the status of tumor-specific CD8+ T cells been detected? If yes, it is necessary to provide related result

Author Response

REVIEWER #1:

1. Manuscript mentioned that “At day 15, chest CT revealed increased circumferential thickness of right pleura and increased amount of pleural effusion.” ”Three months after the surgery, PET-CT revealed local recurrence and systemic metastases at the follow-up visit.””18F-fluorodeoxyglucose-positron emission tomography (FDG- PET) showed high FDG uptake in the thickened right pleura, mediastinal and cervical lymph nodes.” These results are valuable to demonstrate the changed status of patients. The images of these tomography should be added in the manuscript.

    Reply

FDG-PET images have been added.

2. Manuscript also mentioned that “Pathological diagnosis based on microscopic and immunohistochemical findings was poorly differentiated non-small cell carcinoma with sarcomatoid differentiation.” “Histopathological examination of the surgical specimen showed spindle cells and giant cells, which was consistent with pleomorphic carcinoma with no evidence of lymph node metastasis”. The images of immunohistochemical result and Histopathological examination also should be added in the manuscript.

    Reply

Representative histopathological images have been added.

3. Manuscript mentioned that “Contrast CT showed that the tumor diameter increased five times in case 1 and two times in case 2 (Fig. 1–2)”. Tumor should be marked by labels, such as arrows, to make the changed tumor diameter easy to show.

    Reply

We outlined tumor area on CT images.

4. To address TME-related perturbations, immunostaining with PD-1 and CD8 antibodies was performed.” What is reason for detecting CD8+ T cell status? How about detecting the status of CD4+ T cell or CD3+ T cell? Previous studies also mentioned that “ICI therapy resulted in a sustained increase of tumor-specific CD8+ T cells in the blood, which predicted long-term survival“ In the manuscript, Have the status of tumor-specific CD8+ T cells been detected? If yes, it is necessary to provide related result.

    Reply

CD8+ T cells are effector cells regulated by PD-1/PD-L1 signaling. CD3+ and CD4+ cells are not under direct influence of PD-1/PD-L1 axis, and, therefore, we didn’t extend our study to these cell populations.

Re. tumor-specific CD8+ T cells in the blood, such data are not provided in the available clinical records. We did not address this parameter specifically because our study was mainly focused on the tissue level accessible by microscopy.

Reviewer 2 Report

Authors report valuable two cases of NSCLC patients with fatal outcome. I just found minor errors and recommend to revise them as below.

In line 42: There should be full expression of PD-L

In line 146: Foxp3+ CD4+ => Foxp3+CD4+

In line 151: CD8-positive T cells => CD8+ T cells

In line 189, 190, 192, 231 and 232: CD8+ => CD8+

Author Response

REVIEWER #2:

In line 42: There should be full expression of PD-L

In line 146: Foxp3+ CD4+ => Foxp3+CD4+

In line 151: CD8-positive T cells => CD8+ T cells

In line 189, 190, 192, 231 and 232: CD8+ => CD8+

    Reply

All typos have been fixed

Reviewer 3 Report

This study is valuable for reporting changes in the incidence of PD-L1 before and after ICIs.

May be an important clue in elucidating mechanisms of ICI resistance acquisition.

In the present study, the incidence of PD-L1 was relatively low in the viable area and, conversely, the incidence of PD-L1 was relatively high in the perinecrotic area after administration of ICIs. It is intuitively easy to understand that where a reduction in the incidence of PD-L1 leads to ICI resistance and where a further reduction is more ICI resistant and has a higher viability.

As a premise, it should be kept in mind that both of these cases are so-called atypical cases in which tumors are rapidly enlarging after ICI administration. The small sample size does not generalize these changes in the incidence of PD-L1.

Both of the two patients in this study died of tumor growth in a very short period (28 days, 9 days) after ICI administration. The considerably shorter time from dosing to death warrants consideration of the possibility that ICI administration caused a rapid increase in tumors rather than becoming resistant.

There is also a mixture of viable and perinecrotic areas within the tumor, and the different incidence of PD-L1 among them may reflect intratumor heterogeneity. In addition, the influence of factors such as richness of blood flow and tumor outside and inside is also considered, and it is necessary to describe in the macro from which part of the pathological tissue was collected.

Using this study as a foothold, it is hoped that the influence of ICI on cancer can be comprehended in a variety of ways, and that it can be connected to the further clinical effect.

Author Response

REVIEWER #3:

As a premise, it should be kept in mind that both of these cases are so-called atypical cases in which tumors are rapidly enlarging after ICI administration. The small sample size does not generalize these changes in the incidence of PD-L1.

    Reply

We agree with the reviewer that evidence based on two cases is not sufficient for making generalizations and solid conclusions. Our main aim was to bring attention of the wide audience to the phenomenon of alteration of PD-L1 expression after ICI therapy. Hopefully future studies will shed a light on its mechanism and possible contribution to the outcome of ICI therapy.

In addition, the influence of factors such as richness of blood flow and tumor outside and inside is also considered, and it is necessary to describe in the macro from which part of the pathological tissue was collected.

    Reply

We believe there are several mechanisms contributing to the “geographical” partitioning of the tumor (e.g. necrotic zone, perinecrotic zone, viable zone, invasive front, peritumoral non-neoplastic zone), including, of course, blood flow. Such mechanisms were comprehensively outlined elsewhere (PMID 11181773, PMID 26771115, PMID 29988496, and more) and were not discussed in details in our case report.

Re. tissue sampling, we reviewed records of CT-guided biopsy and confirmed that tumor specimens in patient 1 (right posterior peripleural mass) and patient 2 (right hilar mass) matched with those taken at the autopsy and evaluated by PD-L1 IHC.